# Evaluation of snow depth retrievals from ICESat-2 using airborne laser-scanning data

César Deschamps-Berger[1,2], Simon Gascoin[1], David Shean[3], Hannah Besso[3], Ambroise Guiot[1], Juan Ignacio López-Moreno[2]

[1]Centre d'Etudes Spatiales de la Biosphère, CESBIO, Univ. Toulouse, CNES/CNRS/INRAE/IRD/UPS, Toulouse, France.
[2]Instituto Pirenaico de Ecología, Consejo Superior de Investigaciones Científicas (IPE-CSIC), Zaragoza, Spain.
[3]University of Washington, Dept. of Civil and Environmental Engineering, Seattle, WA.

*Correspondence to*: César Deschamps-Berger (cesar.deschamps-berger@csic.es)

**Abstract.** The unprecedented precision of satellite laser altimetry data from the NASA ICESat-2 mission and the increasing availability of high-resolution elevation datasets open new opportunities to measure snow depth in mountains, a critical variable for ecosystem and water resource monitoring. We retrieved snow depth over the upper Tuolumne basin (California, USA) for three years by differencing ICESat-2 ATL06 snow-on elevations and various snow-off digital elevation models. Snow depth derived from ATL06 data only (snow-on and snow-off) offers a poor temporal and spatial coverage, limiting its potential utility. However, using digital terrain model from airborne lidar surveys as snow-off elevation source yielded a snow depth accuracy of ~0.2 m (bias)and precision of ~1 m (random error) across the basin, with an improved precision of 0.5 m for low slopes (<10°), compared to eight reference airborne lidar snow depth maps. Snow depths derived from ICESat-2 ATL06 and a satellite photogrammetry digital elevation model have a larger bias and reduced precision, partly induced by increased errors in forested areas. These various combinations of repeated ICESat-2 snow surface elevation measurements with satellite or airborne products, will enable tailored approaches to map snow depth and estimate water ressource availability in mountainous areas with limited snow depth observations.

## 1 Introduction

Seasonal snow provides fresh water resources to over a billion people globally (Barnett et al., 2005; Sturm et al., 2017). The spatial distribution of the mass of snow on the ground (snow water equivalent, SWE) in snow dominated catchments is key information to predict runoff during the melt season (Freudiger et al., 2017). Yet, direct mapping of the SWE in mountains remains technologically challenging (Dozier et al., 2016). Recent studies have shown that the assimilation of remotely sensed snow depth data is a viable method for estimating SWE spatial distribution (Brauchli et al., 2017; Margulis et al., 2019; Deschamps-Berger et al., 2022). Several methods are now available to map snow depth in mountainous catchments of societal or ecological interest, typically larger than 100 km[2]. Calculating the difference between a snow-on and a snow-off

digital elevation model (DEM) is one of the most straightforward methods. Snow-on and snow-off DEMs can be derived from airborne lidar or photogrammetry with resolution and vertical precision of 10-30 cm (Deems et al., 2013; Bühler et al., 2015). However, these flights are expensive, and repeat snow-on flights are only available in a few basins globally. An alternative to airborne campaigns is to compute DEMs from very-high-resolution stereoscopic satellite images (i.e photogrammetric method). Snow depth maps at a resolution between 2 m and 8 m were produced from images of the Pléiades or WorldView constellations with an uncertainty of ~0.30 m to ~0.70 m (Marti et al., 2016; Shaw et al., 2019; McGrath et al., 2019; Deschamps-Berger et al., 2020; Eberhard et al., 2021). The orbits of these satellites enable the imaging of any region of the Earth's surface (cloud-permitting) but the on-demand tasked acquisition mode results in a discontinuous archive in time and space. Based on a different physical approach, snow depth maps have been retrieved from polarized Sentinel-1 C-band synthetic aperture radar (SAR) backscatter by calibration with snow depth measurements at automatic weather stations (Lievens et al., 2019; Lievens et al., 2022). A single global calibration factor yielded an error of ~2 m (mean absolute error) at 250 m resolution. With the 6 to 12 day revisit of Sentinel-1, this approach provides frequent acquisitions globally at an intermediate spatial resolution. However, this method is not applicable during the melt season when the radar signal is absorbed by the liquid water contained in the snowpack.

Spaceborne lidar missions measure elevation along linear tracks parallel to the satellite orbit. The NASA Ice Cloud and Land Elevation Satellite (ICESat) GLAS instrument was operational from 2003 to 2010 and measured the elevation along a single track every 170 m within a footprint of 70 m. Snow depth could be retrieved from ICESat snow-on observations using a reference airborne lidar snow-off DEM (Treichler et al., 2017). At the footprint scale, the snow depth uncertainty reached an RMSE of 1 m. Due to the sampling structure and the accuracy of ICESat, snow depth data were sparse and not retrieved over slopes greater than 10°. This method was best suited to measure snow depth averaged over seasons and elevation bands, which means a coarsening of the temporal and spatial resolution. Since October 2018, the higher resolution follow-up mission ICESat-2 has provided improved elevation measurements using ATLAS, a photon-counting lidar instrument. The tracks of ICESat-2 consist of three beam pairs, each with a strong and a weak beam, and a cross-track distance of 3.3 km between pairs and 90 m between beams. The photon pulses are spaced by ~0.70 m along-track and illuminate an area of ~11 m in diameter (Markus et al., 2017; Smith et al., 2019) with geolocation accuracy of ~3-4 m (Magruder et al., 2021). However, the  ICESat-2 mission operations were designed to increase the spatial sampling coverage of the tracks for biomass applications in the mid-latitudes. Thus, outside of the polar areas, the tracks are offset to fill coverage gaps and rarely perfectly overlap, which precludes a straightforward approach of retrieving snow depth by differencing snow-on and snow-off elevations along every ICESat-2 reference ground track. The individual photon returns, i.e. the ATL03 products, are processed to provide, for instance, estimates of land ice elevation with a 20 m spacing along track (ATL06) or land surface and forest canopy height at a 100 m spacing (ATL08). Other applications for these products have emerged, including attempts to measure snow depth with ATL08 and ATL06 (Hu et al., 2021; Enderlin et al., 2022). Hu et al. (2021) measured snow depth with ATL08 data at few points (N=16) with slopes lower than 1.5° and snowpack shallower than 0.35 m. They suggested that this product may not be suitable for rugged topography. Enderlin et al. (2022) compared ATL06 and ATL08

elevations with reference DEMs derived from satellite photogrammetry and airborne lidar to increase the number of snow depths retrieved. ATL08 snow depth retrievals were found to be reasonably accurate in regions of low slope, but uncertainties increased in mountainous terrain, as previously found by Hu et al., (2021). However, they concluded that snow depth could be measured in mountainous terrain and over a glacier with ATL06 but lacked distributed validation data to estimate the uncertainty of the retrievals. Considering the current need to measure snow depth in mountains and the increasing availability of high-precision elevation datasets, these approaches seem promising.

In this study, we assessed the uncertainty of different approaches to retrieve seasonal snow depth from the ICESat-2 ATL06 products in complex terrain. More specifically, we studied which type of DEM is required as a snow-off elevation source. To address this question, we explored the ICESat-2 ATL06 dataset over the upper Tuolumne basin where airborne snow depth maps are frequently acquired through the Airborne Snow Observatory (ASO). The ASO program provides 3 m resolution snow depth maps with an uncertainty of ~0.1 m (Currier et al., 2019; Mazzotti et al., 2019). The upper Tuolumne basin is ideal for testing new snow depth detection methods as the acquisitions are repeated every two weeks in the melt-period since 2013. We obtained over 100,000 snow-on points between October 2018 and November 2021 from ICESat-2 ATL06 and compared them with an airborne lidar DEM, a satellite photogrammetry DEM and a global DEM derived from X-band InSAR observations (Copernicus DEM). The snow depth retrievals were evaluated against eight airborne lidar snow depth maps from the ASO. Our objective was to assess the uncertainties of these retrievals, and not to characterize the spatial and temporal variability of the snow depth in the upper Tuolumne. The interested reader will find more information about this topic in other studies (Margulis et al., 2019; Pflug and Lundquist, 2020).

## 2. Study site

The upper Tuolumne river basin is part of the Sierra Nevada mountain range (California, USA) and is contained within Yosemite National Park (Figure 1). It is located above the Hetch Hetchy Reservoir which provides fresh water and produces hydropower for the San Francisco region (Painter et al., 2016). It consists of 1100 km² of montane forests and alpine zones spanning an elevation range of 1200 m to 4200 m. Tree cover is composed of deciduous broadleaf and needleleaf evergreens forests and its density varies greatly within the watershed. More than half of the precipitation of this region range falls as snow (Li et al., 2017; Lahmers et al., 2022) with large year-to-year variations of snow accumulation related to low precipitation during pluriannual droughts or strong precipitation events from atmospheric rivers (Hedrick et al., 2019; Pflug et al., 2022).

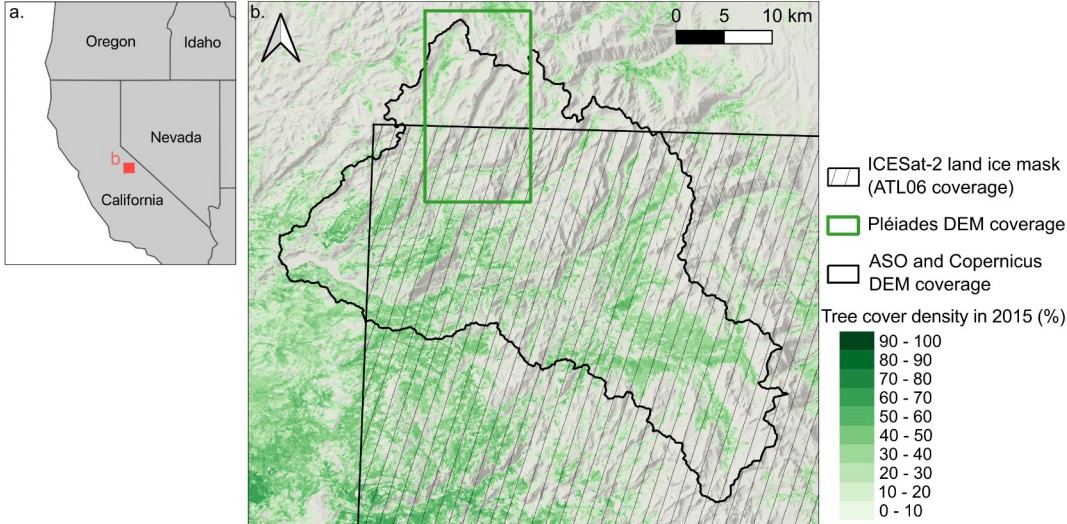

**Figure 1.** The upper Tuolumne basin is located in California, USA (a). The basin is entirely covered by the ASO DEM (black contour) and partially covered by the ATL06 coverage (black hatch) and by the Pléiades DEM (green rectangle). The background map shows a hillshade of the topography and the tree cover density (green shades) (b).

## 3. Materials

### 3.1 ICESat-2 ATL06 elevation product

ATL06 was primarily designed to provide elevation measurements for land ice, yet its coverage extends beyond glacier areas such that ATL06 data are available even in mountain ranges with very limited glacier cover such as the Sierra Nevada (Smith et al., 2019). The ATL06 product is generated by fitting 40 m segments to filtered land-surface photon returns along each of the six tracks, with segments overlapping by 20 m. Photons returned by above ground objects (e.g. vegetation) are included in these fits. The mean surface height of each linear segment is provided as point data positioned at the center of that segment and is labeled $h\_mean$ in the ATL06 data product (Smith et al., 2019). The height $h\_li$ was used as it is calculated after correction of $h\_mean$ for errors in the detection of photons by ATLAS (i.e the transmit-pulse-shape error and the first-photon-bias). The overlap of the segments results in a point located every 20 m along-track for each of the six tracks.

We processed all available ATL06 segments over the upper Tuolumne basin resulting in 265,590 points spanning from 15 October 2018 to 7 November 2021. We excluded segments with errorsgreater than 1000 m (4% of the data), indicated by the field $sigma\_h\_mean$.

### 3.2 Snow-off elevation data

We used three snow-off DEMs from airborne lidar, satellite photogrammetry and satellite InSAR (Table 1):

(i) A digital terrain model (DTM) at 3 m grid spacing was measured with airborne lidar during the ASO campaign on 13 October 2015 (Painter et al., 2016).

(ii) A DEM at 3 m grid spacing was calculated from stereographic images of the satellite Pléiades 1A on 13 August 2017 (Deschamps-Berger et al., 2020). This DEM covers 220 km² of the upper Tuolumne basin (i.e. 20% of the total area).

(iii) A DEM was clipped from the Copernicus-30 global dataset at its native grid spacing of 30 m (COP-DEM-GLO-30-R, https://doi.org/10.5270/ESA-c5d3d65). The Copernicus-30 product was derived from InSAR data of the TanDEM-X mission in most areas, including the upper Tuolumne basin, with some areas filled with miscellaneous external products.

## 3.3. Vegetation and snow-cover products

The Terra MODIS MOD10A1 product was used to retrieve snow cover (Hall and Riggs, 2016; Figure S1). It provides daily snow cover maps with a spatial resolution of approximately 500 m (Hall and Riggs, 2016). The tree cover density was retrieved from the Landsat-MODIS product (Sexton et al., 2013) which provides the proportion of the area occupied by trees at 30 m resolution (Figure 1).

## 4. Methods

### 4.1. ATL06 snow-cover calculation

The number of photons used to calculate the height of each ATL06 segment (*n_fit_photons*) varies with the land cover. In particular, it increases over snow surfaces which are highly reflective at the ATLAS laser wavelength (532 nm). We take advantage of this property to determine the snow presence for every segment. We classified snow as present when *n_fit_photons* exceeded a certain threshold. We determined the threshold by optimizing the accuracy of the classification in comparison to MODIS snow cover data. We first generated a daily gap-free stack of MODIS snow cover area maps by linear interpolation of the normalized difference snow index (NDSI) in the time dimension on a pixel basis followed by a binarization to snow and no-snow using a NDSI threshold of 0.2 (Gascoin et al. 2022). Then, we sampled the MODIS snow maps at each ATL06 segment location for the matching date. The kappa coefficient, a statistic often used to measure the consistency between two classifications (Cohen, 1969), was used to find the optimal threshold, i.e. we determined the threshold which maximized the kappa value by testing all possible values from 0 to 500 photons (Figure S2). This optimization was done separately for the weak (N=123513) and the strong beam segments (N=132289). Figure S1 shows the spatial distribution of the mean annual snow cover duration computed from the interpolated MODIS snow maps over the Tuolumne river basin.

### 4.2 Snow depths calculation

The window used to select the photons and calculate the ATL06 elevations has a maximum length of 40 m and a width
corresponding to the ~11 m ATLAS footprint diameter (Magruder et al., 2021). The ASO and Pléiades DEMs were
resampled from their native resolution of 3 m to 15 m by averaging. The 15 m resolution was selected because (i) it
approximates the spatial window used to calculate each ATL06 segment and (ii) it is an integer multiple of the initial
resolution of the source DEMs. All DEMs were co-registered to the ICESat-2 snow-off point cloud using the Nuth and Kääb
(2011) method. This method relates the horizontal co-registration vector between two elevation datasets with the elevation
difference between the two datasets, the slope and the aspect of the terrain. It can be used with gridded products (e.g. lidar or
photogrammetry DEM) or irregularly distributed points (e.g. ICESat-2 ATL06). The elevation of the raster DEM was
sampled at the ICESat-2 point position with a spline linear interpolation scheme (scipy.interpolate.interp2d). The slope and
aspect were calculated from the DEM and extracted with the same method. The slopes smaller than 10° and steeper than 45°
(empirical thresholds) were excluded to prevent errors in the co-registration vector calculation (Nuth and Kääb, 2011). A co-
160 registration vector was iteratively calculated and applied to the DEM, the aspect and the slope raster. The iteration was
stopped when the co-registration vector was shorter than 0.1 m or when the Normalized Median Absolute Deviation of the
residual (NMAD, i.e. error metric, Höhle and Höhle, 2009) of the elevation difference was improved by less than 1%. After
the horizontal co-registration vector was applied, a vertical shift was applied to the DEM based on the mode of the elevation
residual distribution (Table S1).

Due to the difference in structure between the gridded snow-off DEM and the ICESat-2 snow-on points, the elevation of the
snow-off DEMs was interpolated linearly at each ICESat-2 snow-on point to calculate the "ICESat-2 derived snow depth".
The ICESat-2 derived snow depth products were labelled based on the snow-off DEM source, e.g. "IS2-ASO" refers to the
snow depth computed as the difference between ICESat-2 (IS2) snow-on points and ASO snow-off DEM (Table S2).

**Table 1. Elevation and snow depth dataset used in this study.**

| Data | Source | Structure | Snow cover | Spatial spacing | Date |
|---|---|---|---|---|---|
| Elevation points | ICESat-2 ATL06 | Points | Snow-on and snow-off | 20 m | 2018-10-15 to 2021-11-07 |
| Digital Terrain Model | Airborne lidar (ASO) | Regular grid | Snow-off | 15 m | 2015-10-13 |
| Digital Surface Model | Satellite photogrammetry (Pléiades) | Regular grid | Snow-off | 15 m | 2017-08-13 |
| Digital Surface Model | Copernicus DEM – 30 m | Regular grid | Snow-off | 30 m | 2012 - 2014 |
| Snow depth map | Airborne lidar (ASO) | Regular grid | - | 15 m | 2019-03-24 2019-04-17 2019-05-03 2019-07-05 2020-04-13 2020-05-07 2020-05-22 2021-04-29 |
| Tree cover density | Landsat-MODIS | Regular grid | - | 30 m | 2015 |
| Snow cover | MOD10A1 | Regular grid | - | 500 m | 2018-10-15 to 2021-11-07 |

## 4.3 Evaluation of the snow depth estimates

Eight snow depth maps at 3 m grid spacing from the ASO program were available at different dates over the study period (Table 1). The maps were shifted horizontally by the same translation vector used to co-register the ASO DTM to the ICESat-2 snow-off points. The ASO snow depth maps were also resampled by averaging at 15 m for comparison with the ATL06 points. For each ICESat-2 derived snow depth, the snow depth value of the closest ASO snow depth map in time was extracted. Hereafter, we used the term accuracy or bias to describe systematic errors in snow depth while precision was used for random errors (Hugonnet et al., 2022). The accuracy of the ICESat-2 derived snow depths was evaluated with the median of the residuals (e.g. IS2-ASO snow depth minus ASO snow depth) while the precision was evaluated with the NMAD, a measure of dispersion that is robust to outliers. We analyzed the ICESat-2 derived snow depth from the strong beams, the weak beams, and all beams (strong and weak).

The uncertainty of airborne and satellite laser elevations increases  with surface as steep slopes spread the photons return timing compared to flat terrain (Deems et al., 2013; Treichler et al., 2017; Smith et al., 2019). This holds for photogrammetry derived elevation as well, partly due to the strong distortion of the images in the steep slopes (Berthier et al., 2007; Lacroix, 2016; Shean et al., 2016). Thanks to the spatially dense photon detection of ICESat-2, the uncertainty of ATL06 only increases for slopes greater than 60° (Figure S3). We evaluated the impact of slopes on ICESat-2 derived snow depth using slope maps derived from the ASO DTM. Vegetation type and density (shrubs, isolated trees, forests) is also expected to impact the accuracy and precision of the ICESat-2 derived snow depths as vegetation is handled differently in each elevation source (Deems et al., 2013; Smith et al., 2019; Piermattei et al., 2019). The ICESat-2 ATL06 points were produced without explicitly excluding the photons reflected by the vegetation, thus including photons from the top of the canopy to the ground. The ASO DEM is a DTM, i.e. the ground surface is measured with vegetation excluded. The Pléiades DEM measures the visible surface of the vegetation, i.e. a digital surface model. Therefore, the impact of the vegetation on the ICESat-2 derived snow depths was also evaluated using the tree cover density from the Landsat-MODIS 30 m product (Sexton et al., 2013).

## 5 Results

### 5.1 Spatial and temporal availability of ATL06

Figure 3a shows the 255,802 ATL06 points available over the 1100 km² of the upper Tuolumne river basin between 15 October 2018 and 7 November 2021. The number of photons returned for each ATL06 segment varies seasonally and is lowest from June to October during the snow-free season (Figure 2). The optimization of the photon count threshold gives a clear and unique optimum (Figure S2), with 50 photons for the weak beam segments and 186 photons for the strong beam segments. With these thresholds, 59% of the ATL06 points were classified as snow-off. The remaining snow-on points were distributed over 50 unique dates, with half of these dates containing less than 700 points and the remaining dates with more, up to 8000 points, which means at best a coverage of 1.8 km² at a single date if gridding the points on a 15 m grid. About half of the ATL06 points were in areas with a low tree cover density (< 10%) of which 45% were snow-covered. Some snow-on and snow-off points were obtained in areas with higher tree cover density up to 70%, close to the maximum observed in the upper Tuolumne basin (72%).

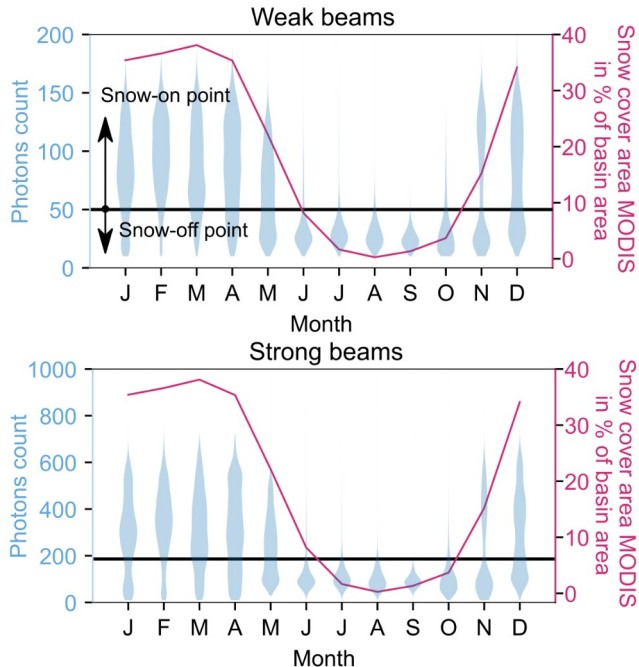

**Figure 2.** Monthly distribution of photons counts by beam type (strong vs. weak) for all ICESat-2 ATL06 points over the upper Tuolumne basin between October 2018 and November 2021 (blue). ICESat-2 has three pairs of beams. Each beam of a pair is either strong or weak depending on the number of photons per pulse. The photon count thresholds (black line) to classify snow-on and snow-off points were optimized with MODIS snow cover. The monthly mean snow cover area (red) from MODIS over the period is shown and a map of the the snow cover duration derived form the MODIS time series is provided in Figure S1.

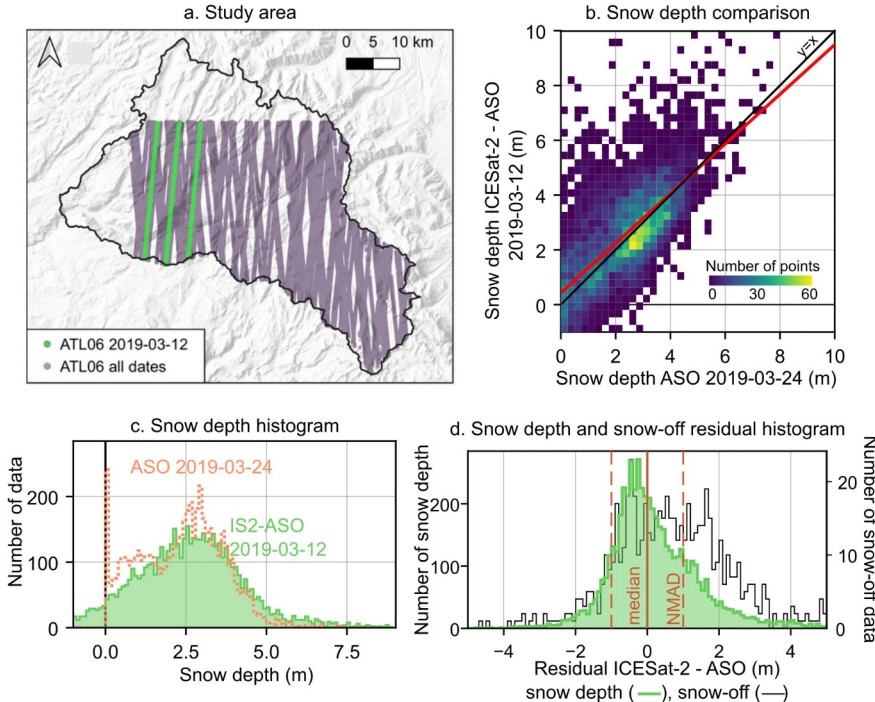

**Figure 3.** (a) Map showing all the ICESat-2 ATL06 points available over the upper Tuolumne basin between October 2018 and November 2021 (purple), with the 12 March 2019 track highlighted (green). (b) Two-dimensional histogram heat-map and (c) general distribution of the ICESat-2-ASO snow depth values on 12 March 2019 (green) and airborne lidar snow depth twelve days later (orange). (d) Histogram of the snow depth residuals (green) and the snow-off residuals (black). Red lines show the median plus/minus the NMAD of the snow depth residual.

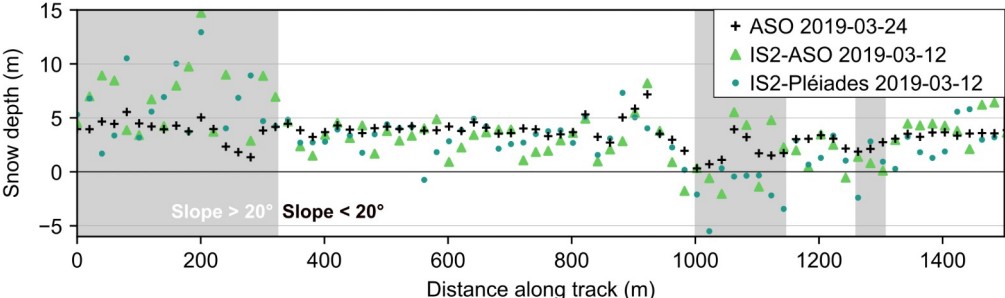

**Figure 4.** Transect of the snow depths on 12 March 2019 derived from ICESat-2 – ASO DTM (green triangles), ICESat-2 – Pléiades DSM (green circles) and on 24 March 2019 by ASO (black cross). Slopes steeper than 20° are shaded as an indication of areas prone to larger errors. This transect is the northernmost of the first beam available on that date (Figure 3a).

**5.2 Impact of the snow-off DEM source**

In the next sections we present results for 12 March 2019 as it is the only date with snow-on points covering a large range of snow depth, which intersect the Pléiades snow-off DEM coverage and with an ASO snow depth map acquired only 12 days later. The snowpack changed a little as the Lower Kibbie Ridge SNOTEL station (2042 m a.s.l., 10 km east of the basin) measured +0.01 m water equivalent (w.e.) change between the ICESat-2 track (12 March) and the ASO snow depth map (24 March).

On 12 March 2019, we obtained the best results from the combination of ICESat-2 ATL06 and ASO snow-off DEM, IS2-ASO (Figure 3, 4, 5, Table S2). The IS2-ASO derived snow depths have a median bias of 0.00 m and a precision of 1.00 m (NMAD). . IS2-Pléiades snow depths have a similar precision (NMAD=1.08 m) but a negative bias (median=-0.53 m). More points were available for IS2-ASO (N=5449) than IS2-Pléiades (N=1295), making the former evaluation more robust, but also possibly impacting differently the uncertainties of each methods on that date (see Sections 5.3 and 5.4). Apparent

negative snow depths in IS2-ASO represent 10% of the snow depths (Figure 3c). These anomalous values are found over shallow snowpack and in areas with slopes greater than 10°. The IS2-Copernicus snow depths showed the worst precision (NMAD=3.00 m) and a low accuracy (median=-0.53 m) (Table S2). Thus, we disqualified the IS2-Copernicus 30 m snow depths and excluded them from subsequent analysis (Figure S4).

The other dates mirror the accuracy and precision found on 12 March 2019 for IS2-ASO (Figure 5d). The NMAD of the
240 snow depth residuals on the eight dates available for evaluation ranges from 0.60 m to 1.16 m, 0.89 m on average. The median of the residuals ranges from -0.65 m to 0.23 m, -0.17 m on average. The two other dates available for the evaluation of IS2-Pléiades show a similar precision with NMAD equal to 1.01 m and 1.16 m while the accuracy was lower with median residuals of -0.68 m and -0.90 m (Figure 5e). The snow-off residuals of IS2-ASO have a lower precision than the snow depths residuals with a NMAD of 1.28 m over all snow-off points (Figure S8, Table S2). The same is observed for the IS2-
245 Pléiades residuals with an NMAD of 1.47 m for all snow-off points.

## 5.3 Impact of the terrain slope

The ICESat-2 derived snow depth showed better agreement with ASO snow depth in areas with low slopes (Figure 5a). For slopes below 10°, IS2-ASO and IS2-Pléiades had a better precision with a NMAD of, respectively, 0.39 m and 0.84 m on 12
March 2019 compared to 1.00 m and 1.08 m for all snow depths on that date. The accuracy for this range of slopes was lower for IS2-ASO product (median=-0.35 m) compared to all the points available that date (median=0.00 m). The IS2-Pléiades accuracy was similar with a median of -0.56 m. The co-registration corrected the vertical bias on all points with slopes up to 45° and cannot ensure a lack of bias for any subset of slopes (e.g. slopes between 0° and 10°).

The IS2-ASO snow depth precision and accuracy worsened with increasing slope. The median residual increased gradually
from -0.35 m for slopes between 0° and 10° to +0.59 m for slopes between 30° and 40° in contrast with the median residual of IS2-Pléiades which decreased in absolute by 0.14 m only from -0.56 m to -0.42 m. Over the same range of slopes, the precision of IS2-ASO decreased as well with the NMAD growing from 0.39 m to 1.48 m. The NMAD of IS2-Pléiades grew comparatively less, from 0.84 m to 1.42 m for the same slopes.

## 5.4 Impact of the vegetation density

The IS2-ASO snow depth accuracy and precision were roughly constant for of tree cover density up to 60%, i.e. the maximum sampled by the 12 March 2019 tracks (Figure 5b). This suggests that ICESat-2 ATL06 points captured the surface elevation below the canopy in this area despite the vegetation. The IS2-Pléiades snow depth was sensitive to the tree density with a decrease in precision and a strong negative bias for tree cover density between 30% and 40% (median=-1.52 m) and between 40% and 50% (median= -4.12 m) compared to the best results measured with tree cover density lower than 10% (median= -0.20 m). The precision was better for areas with low tree cover density (NMAD=0.98 m) compared to areas with tree cover density between 40% and 50% (2.51 m).

## 5.5 Impact of the beam strength

Analysis of the snow-on points from the strong or the weak beam shows lower precision and larger bias for the strong beam on 12 March 2019 for IS2-ASO and IS2-Pléiades (Figure S5 and S6 a to c). The bias of IS2-ASO from the weak beam is smaller than the bias for the strong beam on most dates but the impact of the beams strength on the precision is not systematic (Figure S5 and S6 d and e).

**Snow depth residual (m)**
**ICESat-2 derived snow depth on 12 March 2019 minus ASO snow depth on 24 March 2019**

**Figure 5.** Snow depth residuals (ICESat-2 derived snow depth minus ASO snow depth). Each group of boxplots corresponds to different snow-off DEM source (ASO DTM in green, Pléiades DSM in blue). Within each group, the boxplots are classified by terrain slope (a), tree cover density (b) and snow depth magnitude (c). The snow depth derived from ICESat-2 and the ASO DEM are the most accurate and precise for all tree cover densities. Snow depth residuals when an ASO snow depth map is available at less than 20 days (d, e). The sampling of the breakdown variables differs due to the different coverage of the snow-off DEMs, and transparent boxplots show the data where less than 100 points were available. The black boxplot corresponds to the residual on 12 March 2019 shown in upper panels.

## 6 Discussion

### 6.1. Impact of the snow-off source

The most accurate and precise ICESat-2 snow depths were obtained when using the airborne lidar snow-off DTM (e.g. IS2-ASO). The bias measured were typically 0.20 m in absolute and the precision around 1.20 m or less. The errors in snow depth increase with slope but do not depend on the tree cover density. The airborne lidar DTM measures the ground surface below the tree canopy and ensures ICESat-2 snow depth retrieval even in forest even with tree densities up to 60%, close to the maximum observed in this area. Our results suggest that using a satellite photogrammetry snow-off DEM (e.g. IS2-Pléiades) is a viable alternative in some areas as it provides snow depth with a similar accuracy and precision to airborne lidar for tree cover density below 20% and low slopes. Snow depths derived using the satellite photogrammetry DEM degrade rapidly when tree cover density increases and leads to marked bias. The slope dependence of ICESat-2 snow depth uncertainty varies for airborne lidar and satellite snow-off DEM source, with IS2-ASO showing an increase with slope and IS2-Pléiades showing a constant bias (Figure 5a). This discrepancy between the two DEMs is observed as well even when they are co-registered together (Figure S2 in Deschamps-Berger et al., 2020) but remains unexplained. There is no relationship between the snow depth residual and the time offset between ICESat-2 and ASO acquisition date (Figure 5 d, e). The advantage of combining ICESat-2 with external DEMs to retrieve snow depth compared to times series of DEMs, is that the former method only requires a single DEM to then retrieve snow depth for all subsequent ICESat-2 data which are publicly available. On the contrary, the acquisition of a time series of DEMs requires costly and repeated airborne campaigns (Painter et al., 2016) or commercial satellite tasking (Deschamps-Berger et al., 2022). While airborne lidar datasets are increasingly pubicly available in parts of the world (e.g. in North America and Europe) , coverage is limited for the vast majority of the world's mountains. High-resolution DEMs from satellite photogrammetry are now available in the Arctic (Porter et al., 2018), the Antarctic (Howat et al., 2019) and the Himalayas (Shean, 2017). However, the time stamp is not provided in the mosaiced products and this might hinder the identification of the snow-off from the snow-on pixels. In other areas, commercial satellite stereo images can be acquired on-demand to generate a snow-off DEM. The Copernicus-30 DEM has a global coverage but its uncertainties seem to be disqualifying for this application.

ATL06 snow-off segments might be used as snow-off elevation reference. This would prevent mixing various sources of dataset and rely solely on open data. However, in the three years of the study period, only 2% (25 km²) of this mid-latitude

basin were observed without snow (Figure S11). Assuming the 8.2 km² y$^{-1}$ coverage rate remains steady, more than 50 years will be needed to cover half of the basin. Besides, this rate might decrease in the future as more and more ATL06 segments will be redundant and the proportion of areas seasonally snow-covered to be mapped will increase. Thus, we do not foresee the possibility to map snow depth out of the polar regions with ICESat-2 data only. At best, it might be possible to retrieve snow depth for a subset of points using a method of interpolation at the crossing points of tracks (Moholdt et al., 2010).

More overlapping segments should be available in the Arctic and Antarctica thanks to the repeated orbits in the polar regions.

**6.2 Application to other sites**

The approach described in this article should be transferable in other mountain basins, provided a high-resolution DEM is available. The co-registration parameters and the threshold used to determine ICESat-2 point snow cover classification will likely vary for different sites.

We used the photon counts variable provided with ATL06 segments to determine the snow cover of each segment. It remains uncertain whether the thresholds found here could be transferred in regions with different vegetation cover, terrain roughness

and cloudiness, all of which affect the number of returned photons. In addition, the optimal thresholds for a given region might vary seasonally due to the evolution of the snow albedo and to the vegetation phenology. Further development of this approach could benefit from using higher resolution snow cover maps derived from Sentinel-2 or Landsat images to refine the thresholds or evaluate the snow cover uncertainties (Gascoin et al., 2019).

The horizontal co-registration offsets of the ASO and Copernicus DEMs were small and did not significantly improve the

330 NMAD over the snow-off terrain (Table S1). In contrast, the Pléiades DEM was shifted by 5.63 m which improved the NMAD by 25%. To evaluate the success of the original co-registration method using the ATL06 snow-off points as a reference, we performed a secondary co-registeration of the Copernicus DEM and the Pléiades DEM with the ASO DEM as a reference (Table S3). The small horizontal shift obtained with the second co-registration, with respect to the DEMs resolution, of 0.70 m and 1.38 m respectively for the Pléiades and the Copernicus DEM highlights the good relative

horizontal agreement of the original co-registration. The vertical offsets from the second co-registration were 1.15 m for Pléiades DEM and -0.65 m for the Copernicus DEM, which would impact the accuracy of the snow depths. Thus, it seems preferable to directly co-register the snow-off DEM to the ICESat-2 data as these biases are specific to the elevation sources, in relation with differences in the vegetation measurements and to slope-dependent bias (Figure 5a). The distribution of the elevation difference between ICESat-2 snow-off points and the ASO DTM was positively skewed (Figure S8), suggesting

that vegetation partly led to an overestimation of the ground elevation for snow-off ATL06 points. Acknowledging this, we used the mode of the residual distribution to vertically co-register the ASO DTM. Using the median (Deschamps-Berger et

al., 2020; Shean et al., 2020), would increase the snow depth bias by 0.56 m. The possibility to calculate a single co-registration vector per DEM might depend on the scale of the study site. Here, we were able to successfully map snow depth in areas of 900 km² (intersection of the ASO DEM and ATL06) and 70 km² (intersection of the Pléiades DEM and ATL06). Local co-registration of tiles covering each a quarter of the ASO DEM did not lead to substantial improvement (not shown here). Co-registration of individual ATL06 tracks with airborne lidar or satellite photogrammetry DEMs in Alaska (USA) and Idaho (USA) yielded horizontal shifts in various directions, up to 2.9 m, with no overall systematic shift (Enderlin et al., 2022). The co-registration of individual tracks in the upper Tuolumne basin is not possible on some snow-on dates due to the lack of available snow-off terrain.

## 6.3 Comparison to existing studies

The snow depths derived here from ICESat-2 ATL06 are more accurate, have a finer spatial scale and a denser spatial coverage than snow depths derived with a similar approach from ICESat products (Treichler and Kääb, 2017). ICESat derived snow depths had an RMSE of 1 m over slopes lower than 10° at the 70 m footprint scale (N=27) and steeper slopes were excluded as prone to large errors in ICESat. Here, the IS2-ASO snow depths have an RMSE of 0.85 m (N=907) over slopes lower than 10° on 12 March 2019 at a 3 m scale. The progressive degradation of the accuracy with the increasing slope was also characterized and found to be less pronounced for IS2-Pléiades than IS2-ASO. The rough and vegetated mountain terrain of our study site, as expected, degrades ATL06 accuracy. The accuracy of ATL06 elevations was ten times better over the Antarctic ice sheet than in this study with a precision of 0.09 m (standard deviation) compared to GNSS measurements (Brunt et al., 2019). The calculation of ATL06 elevation from ATL03 products was optimized for land ice which often have flat, smooth and highly reflective surfaces. Improved precision might be obtained tuning the ATL06 parameters (e.g., segment length, photon classification) for mountainous terrain using on-demand processing services such as SlideRule Earth (Shean et al., 2023).

The ICESat-2 ATL06 snow depths (NMAD between 0.5 m and 1.2 m) were less precise than snow depths derived from repeat airborne lidar measurements (Mazzotti et al., 2019) and similar or slightly worse than what was obtained with repeat satellite photogrammetry measurements (Eberhard et al., 2021, Deschamps-Berger et al., 2020; McGrath et al., 2019). In terms of relative error, the snow depth shows a typical error of 40% or less for snow depth greater than 2 m and larger errors for shallow snowpack (Figure S7). This is comparable to the snow depth error from Sentinel-1 retrievals (Lievens et al., 2022). Thus, the existing data assimilation approaches combining satellite photogrammetry or Sentinel-1 snow depth with snowpack models (Shaw et al., 2020, Deschamps-Berger et al., 2022, Alfieri et al., 2022) should be appropriate for ICESat-2 derived snow depth. However, ICESat-2's variable temporal resolution and sparse spatial coverage is unique compared to spatially continuous airborne or satellite maps and gridded snow model results. Figure 6 shows the inter-annual variability of the snow depth gradient with elevation measured by ICESat-2. The ICESat-2 tracks only covers parts of the elevation with snow cover, and the snow depth distribution sometimes differs in both datasets over the sampled altitudes. Estimation of the basin-wide snow volume from ICESat-2 data requires strategies to overcome the spatially discontinuous and variable

sampling of ICESat-2 like extrapolation based on topographic variables (Molotch et al., 2005, McGrath et al., 2018) or through data assimilation (Magnusson et al., 2014, Cluzet et al., 2022). Another promising approach to utilizing ICESat-2-derived snow depth transects comes from Pflug and Lundquist (2020), where characteristic snow patterns in the upper Tuolumne basin were shown to be repeatable and scalable. Small strips of snow depths were matched with a library of distributed snow depth maps from prior years to produce distributed snow depth maps of the basin. An ICESat-2 track might be used in this way to represent a relevant sample of a basin.

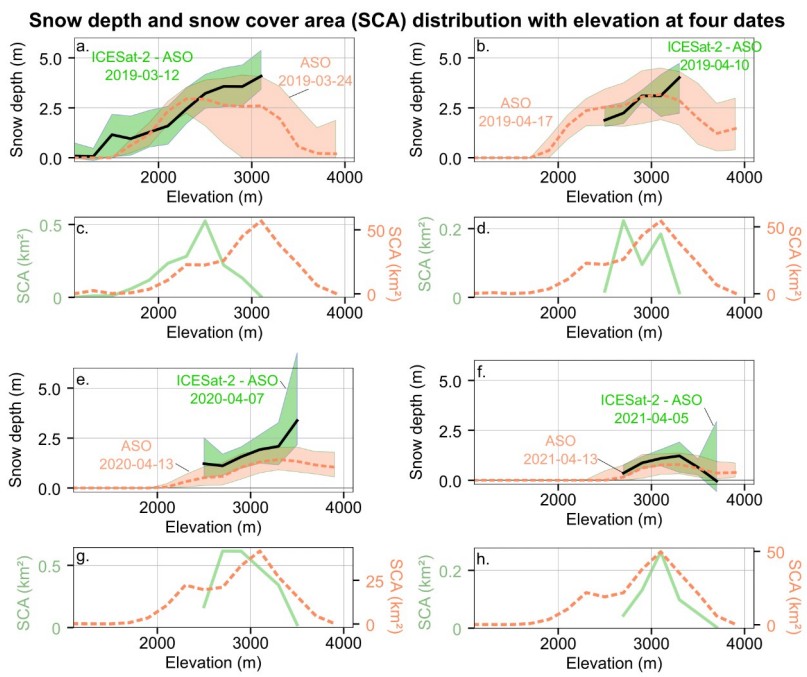

**Figure 6.** Snow depth gradient with elevation (a, b, e, f) from ICESat-2 and ASO snow-off (green) on four selected dates over the three winters of the period and from the closest in time ASO snow depth map (dashed orange). Hypsometry of the snow covered areas (c, d, g, h). The y-axis scale of bottom plots differs to increase the visibility of the smaller surfaces sampled by ICESat-2.

## 7 Conclusion

ICESat-2 ATL06 snow-on elevation combined with airborne lidar or satellite photogrammetry snow-off DEMs offers a promising approach to measure snow depth at high-resolution in mountains. We found that only limited filtering of the standard ATL06 points was required and that a single co-registration with the snow-off DEM was sufficient to obtain usable snow depth measurements. The ATL06 photon counts can be used to classify each point as snow-on or snow-off. By

differencing ICESat-2 snow-on segments with a snow-off airborne lidar DEM, we obtained a precision of ~1 m and a bias of
~0.2 m for a typical mountain environment, i.e. which includes snow depths up to 8 m and a large range of slope. More precise snow depths (~0.5 m) were measured over low slopes (<10°). Similar precision and bias were found for snow depth derived from ICESat-2 and a satellite photogrammetry DEM over low slopes and in terrain with low tree cover density. However, a dense tree cover degraded the snow depth derived from ICESat-2 ATL06 and a digital surface model (i.e. satellite photogrammetry) but had little impact if ATL06 was combined with a digital terrain model (e.g. from airborne lidar). The good quality of the snow depths derived from ATL06 suggests that ATL03 products might provide finer scale and spatially richer snow depth, as each photon returned to ICESat-2 is provided in this product. ICESat-2 ATL06 derived snow depths are a valuable source of information which can be combined with modeling to improve estimates of the amount of water stored in mountains basins, and to understand related spatial and temporal variability. Given the promising results reported here, we suggest that the generation of ATL06 products over non-glacierized mountainous regions is valuable for water resource monitoring in remote mountains across the globe.

**Author contribution**

CDB and SG designed the study. AG performed the initial data curation under and formal analysis further led by CDB and SG. DS and HB contributed to the methodology and the validation of the results. All authors contributed to writing the manuscript. SG, JILM and DS ensured the funding acquisition.

The authors declare that they have no conflict of interest.

**Acknowledgments**

The authors thank Désirée Treichler, Marco Mazzolini  and an anonymous reviewer for their constructive comments. This work has been supported by the Programme National de Télédétection Spatiale (PNTS; grant no. PNTS-2018-4), the Centre National d'Études Spatiales (CNES) and by the Spanish Ministry of Science and Innovation (MARGISNOW project, PID2021-124220OB-100 and HIDROIBERNIEVE project, CGL2017-82216K). AG was supported by Météo-France during the internship which laid the groundwork of this article. DS was supported by NASA award 80NSSC20K0995. HB was supported by NASA award 80NSSC20K1293. We thank Etienne Berthier for reading an advanced version of the manuscript and providing helpful suggestions.

**Open Research**

ICESat-2 ATL06 data were downloaded from https://nsidc.org/data/ATL06/versions/5 on 1 March 2022

ASO snow depth maps and digital surface model were downloaded from https://nsidc.org/data/ASO_3M_SD/versions/1 and https://data.airbornesnowobservatories.com/#

The Pléiades DEM is available at https://zenodo.org/record/6466891#.Yl0SuNPP02w

Copernicus 30 m DEM was downloaded from https://spacedata.copernicus.eu/web/cscda/data-access

Code is available at https://framagit.org/cesardb/icesat-2-mapping-of-snow-height.git

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
