# Peer review of "Evaluation of snow depth retrievals from ICESat-2 using airborne laser-scanning data"

_The Cryosphere, 2022_

## Author Comment (AC1)

**Answer to comments**

**Evaluation of snow depth retrievals from ICESat-2**

**using airborne laser-scanning data**

**tc-2022-191**

We thank the three reviewers for their detailed comments on the article. We took into account several major comments which led to simplifying the manuscript. We removed some parts of the work which did not seem relevant anymore. In detail, we now only present the snow depth derived from the elevation models at 15 m. We removed the snow depth derived from ICESat-2 only data. We also simplified the presentation of the results by only presenting snow depth derived from ATL06 h\_li, excluding the previous comparison between h\_mean and h\_li elevations. We detail these choices and modifications in detail below.

**RC1: 'Comment on tc-2022-191', Anonymous Referee #1, 08 Nov 2022**

There has been increasing interest in using ICESat-2 data for snow depth measurements in mid-latitude locations. This manuscript analyzes the ICESat-2 ATL06 product for its ability to retrieve snow depth in the Tuolumne Basin of California. ATL06 snow-on data is compared to snow-off DEMs generated from ATL06, the Airborne Snow Observatory, Pleiades, and the Copernicus-30 InSAR dataset. The authors found that snow depths were generally most accurate when comparing ICESat-2 to ASO or Pleiades, though accuracy deteriorates with increasing slope.

I do have a concern with the broader narrative that I think needs to be addressed. Early in the text, it is mentioned that current airborne and spaceborne methods of retrieving snow depth suffer from limited spatial and temporal coverage, and ICESat-2 is presented as an alternative. However, the results make it clear that deriving snow depth from only ICESat-2 data is currently impractical, so ICESat-2 snow depth coverage is limited by the same issues as airborne/spaceborne methods. So, my questions are: currently, what are the advantages to using ICESat-2 for snow depth retrievals? Will we get to a point where we can reliably use the "IS2-IS2 DEM" method on a broader scale? I would be very interested to see the authors' viewpoints.

**We now comment on the advantage of this approach compared to the series of DEM approach:**

"The advantage of combining ICESat-2 with external DEMs to retrieve snow depths compared to times series of DEMs, is that the former method only requires a single DEM to then retrieve snow depth for all further ICESat-2 data which are freely available. On the contrary, the acquisition of a time series of DEMs requires costly and repeated airborne campaigns (Painter et al., 2016) or satellite tasking (Deschamps-Berger et al., 2022)."

Agreeing with a comment from reviewer 2, we removed the presentation of the IS2-IS2 results. However, based on our experience, we added a comment on the possibility of using ICESat-2 data only in discussion:

*"ATL06 snow-off segments might be used as snow-off elevation reference. This would prevent mixing various sources of dataset and allow relying solely on free, open access data. However, in the three years of the study period, only 2% (25 km2) of this mid-latitude*

basin were observed without snow (Figure S11). Assuming the 8.2 km2 y-1 coverage rate remains steady, more than 50 years will be needed to cover half of the basin. Besides, this rate might decrease in the future as more and more ATL06 segments will be redundant and the proportion of areas seasonally snow-covered to be mapped will increase. Thus, we do not foresee the possibility to map snow depth out of the polar regions with ICESat-2 data only. At best, it might be possible to retrieve snow depth at a few points using a method of interpolation at the crossing points of tracks (Moholdt et al., 2010). More overlapping segments should be available in the Arctic and Antarctica thanks to the repeated orbits in the polar regions."

**Figure S11.** Surface without snow available in ATL06 since the start of the acquisition (blue) and the related linear regression (red). Each segment of ATL06 is assumed to cover a square of 15 m by 15 m. Segments are assumed to not overlap which is an acceptable assumption for the first years of acquisitions.

Another concern: A paper was recently published by Enderlin et al., (2022) that covers similar topics to this study. Namely, they conduct an investigation of ATL06/08 data over mountainous regions, and they found snow depth uncertainties similar to this study. I do not think a full comparison is necessary, but it would be important to highlight how this study differs from Enderlin et al., (2022) (or builds upon it).

We read with interest the paper published by Enderlin et al. (2022) which shares similar methodology with our work. From our understanding, they could not lead a robust estimation of the uncertainty of the snow depth retrievals due to a lack of validation data. We now discuss this in the introduction (see below) and relate their findings to ours in discussion:

"Other applications have emerged, including attempts to measure snow depth with ATL08 and ATL06 (Hu et al., 2021; Enderlin et al., 2022). Hu et al. (2021) measured snow depth with ATL08 data at few points (N=16) with slopes lower than 1.5° and snowpack shallower than 0.35 m. They suggested that this product may not be suitable for rugged topography. Enderlin et al. (2022) compared ATL06 and ATL08 elevations with reference DEMs derived from satellite photogrammetry and airborne lidar to increase the number of snow depths retrieved. ATL08 snow depth retrievals were found to be hardly reliable in mountainous terrain, in agreement with Hu et al. (2021). However, they concluded that snow depth could be measured in mountainous terrain and over a glacier with ATL06 but lacked distributed validation data to estimate the uncertainty of the retrievals."

**Minor Comments**

Page 1, Line 13: "...and various snow-off elevation sources, including ATL06 snow-off data and external digital elevation models."

This sentence was modified.

**Page 1, Line 26: "catchment" --> "catchments"**

Modified.

Page 2, Lines 61-62: I suggest combining the last two sentences of this paragraph.

Modified these two sentences for:

"However, the orbit of ICESat-2 was designed to increase the spatial density of the tracks coverage for biomass applications in the mid-latitudes. Thus, outside of the polar areas, the tracks are offset and rarely perfectly overlap, which precludes a straightforward approach of retrieving snow depth by differencing snow on and snow off elevations along every ICESat-2 transect."

**Page 2, Line 64: "...from the ICESat-2 ATL06 product"**

This sentence was deleted.

**Page 3, Line 77: What do we mean by "sparse"? For instance, could we reliably obtain ATL06 data from mid-latitude snow catchments?**

This term was confusing in this sentence. We deleted it. We now more clearly comment on ATL06 availability in glacierized areas. Availability of the data for a mid-latitude catchment is answered in the first comment of this review.

"ATL06 was primarily designed to provide elevation measurements on land ice, yet its coverage extends beyond glacier areas such that ATL06 data are available even in mountain ranges with very limited glacier cover such as the Sierra Nevada (Smith et al., 2019)."

Page 4, Lines 117-118: I assume this is supposed to be the ICESat-2 elevation, but I still suggest being specific. Also, why was a cutoff of 45° selected?

This threshold was empirically determined. We changed the formulation:

"The elevation of the DEM is extracted at the ICESat-2 point position with a spline linear interpolation scheme (scipy.interpolate.interp2d). The slope and aspect are calculated from the DEM and extracted with the same method. The slopes smaller than 10° and steeper than 45° (empirical thresholds) are excluded to prevent errors in the co-registration vector calculation (Nuth and Kääb, 2011)."

Page 4, Line 123: Just for clarity, the vertical shift is applied to the DEMs, not the ATL06 data?

Indeed. We added in the text:

"After the horizontal co-registration vector is applied, a vertical shift is applied **to the DEM** based on the mode of the elevation residual distribution (Table S1)."

Page 5, Line 139: NMAD is first introduced on Line 121, so I suggest moving this citation there.

We moved the reference (Höhle and Höhle, 2009) where NMAD is first introduced.

Page 5, Lines 141-144: Is it known if the DEM sources (ASO, Pleiades) have uncertainties over slopes and vegetation? If so, then I suggest mentioning them here.

We completed this part to mention the error in the Pléiades DEM on steep slopes and cite the articles which explain how vegetation is treated in the various dataset:

"The uncertainty of airborne and satellite laser elevations increases when the slope increases as steep slopes spread the photons return timing compared to flat terrain (Deems et al., 2013; Treichler et al., 2017). This holds for photogrammetry derived elevation as well, due to the strong distortion of the images in the steep slopes (Berthier et al., 2007; Lacroix, 2016). Thanks to the spatially dense photon detection of ICESat-2, the uncertainty of ATL06 only increases for slopes greater than 60° (Figure S2). We evaluate the impact of slopes on ICESat-2 derived snow-depth thanks to slope maps derived from the ASO DTM. Vegetation (bushes, isolated trees, forests) is also expected to impact the accuracy and precision of the ICESat-2 derived snow depths as vegetation is handled differently in each elevation source (Deems et al., 2013; Smith et al., 2019; Piermattei et al., 2019)."

**Page 6, Line 158: "photon" --> "photons"**

Modified.

Page 7, Lines 165-166: It would be nice to have a figure that shows the tree cover density across the region of interest.

We modified the figures so that Figure 1 presents the study area and shows the tree cover density:

---

## Author Comment (AC2)

**Evaluation of snow depth retrievals from ICESat-2**

**using airborne laser-scanning data**

**tc-2022-191**

*We thank the three reviewers for their detailed comments on the article. We took into account several major comments which led to simplifying the manuscript. We removed some parts of the work which did not seem relevant anymore. In detail, we now only present the snow depth derived from the elevation models at 15 m. We removed the snow depth derived from ICESat-2 only data. We also simplified the presentation of the results by only presenting snow depth derived from ATL06 h_li, excluding the previous comparison between h_mean and h_li elevations. We detail these choices and modifications in detail below.*

**RC2**: 'Comment on tc-2022-191', Désirée Treichler, 21 Nov 2022

The authors present a study where they assess the potential of spaceborne laser altimetry from ICESat-2 to measure snow depth when compared to snow-free digital elevation models (DEMs) from various sources. These are compared to snow depth maps from airborne lidar data acquired close in time as a verification of their results. The study focuses on an extensive analysis of the residuals between the two datasets including considerations about spatial resolution, forest cover and surface slope. Accurate methods to measure snow depth from space would meet a great need within snow science. This study thus has the potential to be very useful for the snow community.

However, the manuscript currently has two main issues (outlined below) that limit its usefulness: 1.) With the focus on residuals, there are barely any actual snow depth results - which makes it rather too difficult for potential users of this novel data/approach to know whether this is useful for their purpose; and 2.) some of the methods/processing developed for the ICESat-2 approach leave a somewhat unfinished impression, i.e. some changes may greatly improve the quality of the snow depth measurements from ICESat-2. I suggest that the authors review their article with a focus on these two aspects.

This review was done in collaboration with a Ph.D. candidate in training who is specialising in this field.

Main/general comments:
* * *
1.) referring to the lack of snow depth results: As a reader, I would have expected an analysis of the actual snow depth measurements retrieved from ICESat-2 data, not only an analysis of the residuals with the validation dataset(s) where the snow depths themselves are no longer visible. For example, users may want to know whether a single ICESat-2 ground track crossing their catchment can give them any useful snow depth data. You could thus provide a plot of a snow depth transect, or plot the snow depths of one overpass on a map. Given that individual measurements rather seem too inaccurate to be useful, users may wonder whether the ATL06-derived data accurately reproduces how snow depths vary in elevation or for different aspects. This would provide a much more approachable basis for

the thorough analysis of the residuals that you already have.

*We had to keep balance between this comment and request from Reviewer 1 to move figures from supplement to the main text which would have increased the share of the article dedicated to residuals. Following this suggestion, we propose to add a snow depth transect in Fig. 4 (see below). As suggested, snow depth variations with elevation are already presented in supplement (Fig S12). We also propose to move it in the main manuscript to enhance its visibility.*

[Figure]

**Figure 4.** *Transect of the snow depths on 12 March 2019 derived from ICESat-2 – ASO 3 m (green triangles), ICESat-2 – Pléiades 3 m (green circles) and on 24 March 2019 by the ASO (black cross). Slopes steeper than 20° are marked on the X-axis as an indication of areas prone to errors. This transect is the northernmost of the first beam available on that date (Figure 3a).*

[Figure]

*Figure S12 (to be moved to the manuscript). Snow depth gradient with elevation (top) from ICESat-2 and ASO snow-off (green) on four dates over the three winters of the period and from the closest in time ASO snow depth map (orange). Hypsometry of the snow covered areas is shown below. Note the different y-axis scale to increase the visibility of the smaller surfaces sampled by ICESat-2.*

*We think that analyzing the residuals is informative and will serve future application of the method. We believe that the title of the article is clear about the focus of the work but we are open to suggestions that would prevent confusion. We added a sentence in the introduction :*

*"Our objective was to assess the uncertainties of these retrievals, and not to characterize the spatial and temporal variability of the snow depth in the upper Tuolumne. The interested reader will find more information about this topic in other studies (Margulis et al., 2019; Pflug et al., 2020)."*

2.) referring to methods/processing, this includes several points:

- from Fig. S2 it seems that coregistration between the DEMs and ICESat-2 was incomplete, with the remaining shifts causing elevation errors that are far larger than the assumed snow depths.

*On Figure S2, we showed that the co-registration of the Copernicus 30 m DEM on ICESat-2 was incomplete. Working on this comment helped us to find a bug in our workflow. The corner or the center of the pixels were alternatively taken as the pixels coordinate in the co-registration routine and the elevation extraction. This changed the co-registration vectors but had little impact on the snow depth uncertainty. It improved the co-registration of the Copernicus 30 m DEM to the ICESat-2 snow-off points. We updated all comments related to the co-registration.*

*To ensure the performance of the co-registration, we co-registered the Pléiades DEM and Copernicus DEM to the ASO DEM after their respective co-registration to ICESat-2 (Table S3). This results in small horizontal co-registration vectors (compared to the DEMs resolution) of 0.70 m for the Pléiades DEM and 1.38 m for the Copernicus DEM and little improvement of the NMAD of the residual. This shows the good relative agreement between the co-registration of the ASO and Pléiades DEMs to ICESat-2. We consider that the absolute accuracy of the co-registration is estimated with the different snow depth residuals presented in the article.*

*"The horizontal co-registration of the ASO and Copernicus DEMs were small and did not significantly improve the NMAD over the snow-off terrain (Table S1). The Pléiades DEM was shifted by 5.63 m by the co-registration which improved the NMAD by 25%. We co-registered the Copernicus DEM and the Pléiades DEM to the ASO DEM after co-registration to the ATL06 snow-off points to evaluate the success of the co-registration processes (Table S3). The small residual shift, with respect to the DEMs resolution, of 0.70 m and 1.38 m respectively for the Pléiades and the Copernicus DEM highlights the good relative agreement of the co-registration. The vertical co-registration are significant with 1.15 m for Pléiades DEM and -0.65 m for the Copernicus DEM and could lead, if applied, to changes in the accuracy of the snow depths. It seems preferable to co-register the snow-off DEM to the ICESat-2 data as these biases are specific to the elevation sources, in relation with differences in the vegetation measurements and to the slope-related bias (Figure 5a)."*

The ASO DEM seems to be the most carefully curated and accurate reference DEM in the area and may be the better coregistration basis than ICESat-2.

*We understand that it is suggested to co-register the DEMs to the ASO DEM since it has the best quality among the dataset we used. This does not seem necessary since the answer to the previous comment shows that the horizontal shift would be small. Besides, this approach would cancel the main advantage of the Copernicus DEM which is to be freely and globally available and of satellite photogrammetry DEMs which can be acquired worldwide.*

Given that all elevation datasets are patched together from several individual acquisitions, your results may improve greatly if you coregister smaller spatial tiles/units individually.

*Following this suggestion, we splitted the ICESat-2 snow-off point and the ASO 3 m DEM in four tiles (see figure below) and co-registered each tile of the DEM individually. The co-registration vector of the tiles differ from the one calculated for the whole area by as much as 0.54 m but the relative gain in NMAD of the snow-off residuals remain similar, around 10%.*

[Figure]

*Figure. Left: map showing the ATL06 segments split in four tiles (north-west, north-east, south-east and south-west). Middle: horizontal co-registration vector for each tile of the ASO 3 m DEM (color) and for the whole DEM (black). Right: mode of the elevation residual after horizontal co-registration.*

*We then merged the four individually co-registered DEM and calculated the snow depth and the associated residuals (see boxplot figure below). We do not observe clear differences between the two co-registration methods. We conclude from this experience that similar results can be obtained at the scale of the whole basin (~1000 km²) or at the scale of a tile*

[Figure]

*(~250 km²).*

*Figure (not added in the manuscript). Distribution of the residuals of the IS2-ASO 3 m snow depth either co-registered as a whole (i.e. as described in the manuscript, green boxes) or co-registered by tile (i.e. as described above, white boxes).*

*As suggested further in this review, we applied the same tiling approach with the Copernicus 30 m DEM with similar results to the ASO tiling approach which means no significant improvement. We did not apply this method to the Pléiades DEM as it has a smaller area.*

[Figure]

*Figure. Left: horizontal co-registration vector for each tile of the Copernicus 30 m DEM (color) and for the whole DEM (black). Right: mode of the elevation residual after horizontal co-registration.*

[Figure]

*Figure (not added in the manuscript). Distribution of the residuals of the IS2-Copernicus 30 m snow depth either co-registered as a whole (i.e. as described in the manuscript, purple boxes) or co-registered by tile (i.e. as described above, white boxes).*

*We now comment on this verification in discussion:*

*"The possibility to calculate a single co-registration vector per DEM might depend on the scale of the study site. Here, we were able to successfully map snow depth in areas of 900 km² (intersection of the ASO DEM and ATL06) and 70 km² (intersection of the Pléiades DEM and ATL06). Refined co-registration of tiles covering each a quarter of the ASO DEM did not lead to substantial improvement (not shown here)."*

You only analyse the residuals of the snow-on data points - how does the distribution look

like for the snow-off data? Similar, narrower, wider? Do you see the same bias for snow-off points and snow-on points from the same overpass (if there are overpasses with both snow-on and snow-off data)?

*Snow-off points are indeed not available at all dates (e.g. no snow-off points for Pléiades on 12 March 2019). The width of the snow-off distribution is described by the NMAD of the snow-off distribution, provided in Table S1. We added a figure showing the distribution of all snow-off points (Figure S8), a figure showing the distribution of the snow depth residuals and of the stable terrain by dates (Figure S9) and we added a line showing the distribution of the snow-off points on 12 March 2019 for IS2-ASO 3 m (Figure 3d).*

*We also comment on the snow-off distribution:*

[Figure]

**Figure S8.** *Distribution of the snow-off residual (ICESat-2 minus DEM) before (black line) and after coregistration (color) for the ASO DTM (left), the Pléiades DEM (middle) and the Copernicus DEM (right). The vertical lines show the median plus/minus the NMAD after co-registration.*

[Figure]

**Figure S9.** *Snow depth (colored boxes) and snow-off (white boxes) residuals for ICESat-2 – ASO (left) and ICESat-2 – Pléiades (right). Transparent boxplots show the data where less than 100 points were available.*

- ATL06 data is an averaged product of 40m long track segments with photon returns spread ca. 11m in width (across-track). It assumes flat glacier surfaces, as you write. Although the segments are overlapping, resulting in 20m spatial sampling, I find it problematic to translate this into "20m resolution" given the potentially strong terrain differences within the 40x11m segments. You analyse the effect of the resolution of your snow-off DEMs, and by this you mean the cell size of the pixel the ATL06 centre data point happens to fall on - but one would assume that a correct representation of the ATL06 footprint including all DEM cells would make a much greater difference for the resulting snow depths, especially with a high-resolution snow-off DEM such as the ASO DEM? When analysing the effect of resolution, I would thus expect that you rather assess/also include the latter approach. In my opinion, this would strengthen the usefulness of your resolution analysis considerably.

*We paid attention to not describe the ATL06 as a 20 m resolution product to keep the distinction between the sampling distance and the resolution. Following this comment, we reconsidered our approach and concluded that the use of the snow-off DEMs at high-resolution (3 m) was not justified. We removed these results from the article (i.e. IS2-ASO 3 m, IS2-Pléiades 3 m).*

*We agree that the scale of an ATL06 segment is a rectangle of 40 m by 11 m. We understand that it would be interesting to include all the pixels located in this rectangle as done, for instance, by Enderlin et al. (2022). However this requires a dedicated algorithm which must calculate the corners of the rectangle associated with each segment. We believe that using an intermediate DEM resolution provides a simpler but worthy alternative, with the asset of relying on common tools (i.e. gdalwarp). We explain this in the methods:*

*"The ASO and Pléiades DEMs were resampled from their native resolution of 3 m to 15 m by averaging. The 15 m resolution was selected to match as much as possible the window used to calculate each ATL06 and to be a multiple of the initial resolution of the DEMs. The window used to select the photons and calculate the ATL06 has a maximum length of 40 m and a width corresponding to the ATLAS footprint that is 11 m."*

- Among other methods you also assess the quality of snow depths derived from ICESat-2 data only, i.e. snow-on minus snow-off tracks. Not depending on other data sources would obviously be very useful but I have great doubts whether the approach that you chose in this study is sound, given the rough terrain in your study area. The ATL06 data are already spatially averaged over a 40m segment, and by converting the ICESat-2 data to a grid with a nearest neighbour approach, you effectively shift the ATL06 data around in space. To get snow depths, you again use a nearest neighbour approach (L126) with the same effect. The sum of this results in very high elevation uncertainty. The approach also seems to multiply the area in which ICESat-2 provides snow-off data by a factor of ten (25 km2 snow-off data from 59% of the points, 1.8 km2 snow data from 41% of the points), which is puzzling. This approach may be justifiable in polar areas (where tracks are repeated nearly exactly) and possibly also flat prairie landscapes but it performs very poorly in your area, as you show yourself, making it not convincing. It thus rather weakens the impression of the study as a whole and you may consider removing this part in this study (and rather publish it for a study area where it can be better justified?). If you want to include ICESat-2-only snow depths you could instead focus on cross-point comparisons for snow-off and snow-on tracks, as has been done extensively for ice sheet/glacier changes for the predecessor ICESat data. Repeating that approach would be easy to justify and testing it may be appreciated by the community.

*We fully agree with this comment. We decided to apply this simple method at the beginning of the work and failed to reconsider it objectively along the way. We removed it from the*

*article and now only comment on the potential of IS2 alone snow depth in discussion:*

*"ATL06 snow-off segments might be used as snow-off elevation reference. This would prevent mixing various sources of dataset and allow to rely solely on free, open access data. However, in the three years of the study period, only 2% (25 km²) of this mid-latitude basin were observed without snow (Figure S11). Assuming the 8.2 km² y-1 coverage rate remains steady, more than 50 years will be needed to only cover half of the basin. Besides, this rate might decrease in the future as more and more ATL06 segments will be redundant and the proportion of areas seasonally snow-covered to be mapped will increase. Thus, we do not foresee the possibility to map snow depth out of the polar regions with ICESat-2 data only. At best, it might be possible to retrieve snow depth at a few points using a method of interpolation at the crossing points of tracks (Moholdt et al., 2010). More overlapping segments should be available in the Arctic and Antarctica thanks to the repeated orbits in the polar regions."*

*About the increase of area. Our phrasing was not clear, we meant that 1.8 km² at best can be obtained for a single date, not for the whole of the snow-on segments available. We thus changed this sentence:*

*"The remaining snow-on points were distributed on 50 dates with half of the dates containing less than 700 points and the remaining dates with more, up to 8000 points, which means at best a coverage of 1.8 km² **at a single date** if gridding the points on a 15 m grid."*

Minor/specific comments:
* * *
P1, L15: you choose the terms accuracy and precision for bias/uncertainty in your document. That's OK, but note that this may be less intuitive for some readers compared to other terms (uncertainty, bias). Please ensure that this is consistent/translatable throughout your document, e.g.

- mean absolute error for other datasets (L42f)

- bias / precision for your results (L184)

*We now precise the terms we use:*

*"We use the term accuracy **or bias** to describe systematic errors in snow depth while precision is used for random errors (Hugonnet et al., 2022)"*

*The mean absolute error is the metric provided in Lievens et al. (2022) to describe the error, it includes accuracy and precision.*

Introduction

--

P1,L20: seasonal snow / snow pack / snow mass (not cover)

*We changed to "seasonal snow". We note however that "snow cover" could be used as well, as it is defined as "the accumulation of snow on the ground surface" (NSIDC,*

*https://nsidc.org/learn/cryosphere-glossary/snow-cover). This is still in debate in the snow community (https://twitter.com/smlmrn/status/1363565259865391105).*

P1, L23ff: Consider rewriting, as it is not clear from the context where the mentioned assimilated remotely sensed snow depths were coming from, and the second part of the sentence could be more logically connected.

*We simplified this sentence:*

*"Recent studies have shown that the assimilation of remotely sensed snow depth data is a viable method for estimating SWE spatial distribution (Brauchli et al., 2017; Margulis et al., 2019; Deschamps-Berger et al., 2022)."*

P1, L26: Are these global efforts? Or US-focused? unclear

*We deleted this sentence.*

P1f, L29ff: Starting from here, the entire introduction of DEMs and how to translate them into snow depths would benefit from a more structured/complete approach to improve reading flow. For example, you could first introduce all the methods/carriers and then move on to examples (like the ASO). Consider:

*We changed the structure of the introduction. In particular, we moved the ASO part further down the intro to keep the first paragraphs dedicated to the different methods.*

- accurate DEMs can also be acquired from satellite data (you mention only airborne) or drones. Satellites pop up later (L36f) only.

*We kept it this way but we believe it has improved now that the ASO description has been moved out of this paragraph.We do not mention drones as we list the methods adapted for large study sites, i.e. at least 100 km².*

- make sure the readers understand that the Airborne Snow Observatory is a US thing, not a globally available dataset.

*This sentence was deleted.*

- not all readers may know that you essentially mean the same thing with photogrammetry (L29) and stereoscopy(L36)

*We added:*

*"An alternative to airborne campaigns is to compute DEMs from very-high-resolution stereoscopic satellite images (i.e photogrammetric method)."*

P2, L40ff: this is a very different method, consider a new paragraph. Rather use the term backscatter, not observations.

*We would rather not devote a paragraph to this part as i) it is not further used in the article and ii) it fits well in the paragraph about the different methods to map snow depth. However, we now state that this is, indeed, a very different method:*

*"**Based on a different physical approach**, snow depth maps have been retrieved from Sentinel-1 **backscatters** by calibration with snow depth measurements at automatic weather stations (Lievens et al., 2019; Lievens et al., 2022)."*

P2, L48ff: I suggest you add that the method by Treichler & Kääb (2017) required both spatial and temporal averaging, i.e., ICESat was best suited for average winter snow depths over several years.

*We added this:*

*"This method was best suited to measure snow depth averaged over seasons and elevation bands, which means a coarsening of the temporal and spatial resolution."*

P2, L52ff: Consider re-structuring the introduction of ICESat-2 & specs and the level of detail that is suitable here or in a later section where you also introduce sensor/data parameters. You may also consider pointing to a figure that shows the ground track layout/availability (could be a map like in fig. 1). Some suggestions:

- L53: ...a strong and a weak beam EACH…

*We modified this sentence as suggested.*

- L56: 20 m spacing, but 40 m spatial resolution - an important difference for rugged terrain

*We rather agree with this point since we indeed stated the spacing and not the resolution. We would rather not mention the resolution here since due to the width of the track (~11 m), the resolution is not 40 m. This is further discussed in this review and in the article.*

- ATL08 includes land surface and canopy heights (both)

*True. We modified the sentence to state that this list is not comprehensive :*

*"The individual photon returns, i.e. the raw products, are processed to provide, **for instance,** estimates of land ice elevation changes with a 20 m spacing along track (ATL06) or forest canopy height at a 100 m spacing (ATL08)"*

- L60ff: the logical connection is unclear. I recommend to introduce the strategic off-pointing

of ICESat-2 ground tracks earlier (together with the beam design): be very clear that the tracks are not repeated exactly, on purpose. Consider that most readers may not be used to the concept of profile data rather than spatially extensive maps/imagery, and they may also not find it obvious why you need a reference snow-off DEM.

*These sentences were moved while rewriting the introduction.*

P2, L63: New topic, start with "In this study" or something to help the reader.

*We modified this sentence to include this suggestion.*

P3, L67: unclear where the upper Tuolomne basin is -> map figure (see other comments)

*A Figure 1 showing the Upper Tuolumne basin position was added.*

Materials and Methods

- Data, study site and processing steps are a bit mixed in this section. I suggest you try to re-structure this and reconsider the sub-section titles so that they are representative for the respective contents. For example, it could be useful to introduce all important aspects/parameters of the ICESat-2 data first - and possibly all other datasets, too - and move all processing steps (L106ff) to a different sub-section.

*We added a dedicated study site section. We also separated and structured better the data section (ICESat-2 ATL06 elevation product, Snow-off elevation data, Vegetation and snow-cover products) and the methods section (ATL06 snow-cover calculation, Snow depths calculation, Evaluation of the snow depths).*

- It would be very useful to add a map of the study area where you could include standard map layers and coordinates to show the location of your study site, the extents of all your datasets (now missing especially for Pléiades, L102), forest cover, elevations etc.

*We added such a map as Figure 1.*

You may want to embed this in a paragraph where you describe the study site (now spread across introduction and methods). The one sentence at L82f currently lacks context.

*We added a dedicated study site section.*

*"2. Study site*

*The upper Tuolumne river basin is part of the Sierra Nevada mountain range (California, USA) and is contained within Yosemite National Park (Figure 1). It consists of 1100 km² of montane forests and alpine zones spanning an elevation range of 1200 m to 4200 m. Tree cover and terrain slope vary greatly within the watershed. More than half of the precipitation*

*of this region range falls as snow (Li et al., 2017) with large year-to-year variations of snow accumulation (Pflug et al., 2022)."*

P3, L75f: just somewhat beyond - or is it globally available? Why "sparse" - more sparse than elsewhere? Do you mean mountain "ranges"?

*We deleted "sparse" and added "mountain".*

P3, L78f: 40m LINEAR segments? "land-surface photon returns": you may want to specifically mention that "land-surface" includes vegetation/forest photons as these will bias the mean elevation within the segment - something you later address in your forest analyses.

*We added:*

*"Photons returned by above ground objects (e.g. vegetation) are included."*

P3, L83: "granules" unclear

*We replaced "granules" with "segments" which is the term used in the ICESat-2 ATL06 documentation.*

P3, L85: Explain what "sigma_h_mean" and "n_fit_photons" represent and where you get these parameters from

*We modified this part:*

*"We excluded segments with large errors, indicated by the field sigma_h_mean. Segments with errors greater than 1000 m were discarded (4% of the data). The number of photons used to calculate the height of each segment is provided in the field n_fit_photons."*

Snow classification

P3, L86ff: "...as snow and ice are highly reflective...": not a logical continuation of the sentence, needs a clear explanation. You may link this to the strong/weak beams mentioned earlier. Your snow on/off classification is a smart idea but the description here needs to become clearer.

*We appreciate the positive comment on our snow classification method. Following these suggestions, we have clarified the corresponding section as explained below.*

- number of photons - per what?

*We refer to the number of photons used to determine the height of each segment as provided in the ATL06 product. We now explain it:*

*"The number of photons used to calculate the height of each segment is provided in the field n_fit_photons."*

- source/reference and resolution (spatial/temporal) of this dataset are missing. Please introduce it properly.

*We have included the MODIS dataset reference and resolution in the dedicated sub-section of the data section.*

*"The Terra MODIS MOD10A1 product was used to retrieve snow cover (Hall and Riggs, 2016; Figure S1). It provides daily snow cover maps with a spatial resolution of approximately 500 m (Hall and Riggs, 2016). The tree cover density was retrieved from the Landsat-MODIS product (Sexton et al., 2013) which provides the percentage occupied by trees at 30 m resolution (Figure 1)."*

- explain what NDSI means, and justify your chosen threshold

*We have added the meaning of the NDSI. The threshold is the same as Gascoin et al. (2022) (new reference added):*

*"From this product, we generated a daily gap-free stack of MODIS snow cover area maps by linear interpolation of the normalized difference snow index (NDSI) in the time dimension on a pixel basis followed by a binarization to snow and no-snow using a NDSI threshold of 0.2 (Gascoin et al. 2022)."*

- Please explain more clearly how you get from the photon and MODIS data to the photon threshold through Cohen's kappa, the readers probably don't know this approach. The curves in the supplementary don't help much there (how were they made? Adding a scatter plot with the number of photons vs MODIS fSCA may help).

*Here we compare the photon counts with a binary variable (snow/no snow, not fSCA) hence a scatterplot would not really make sense. To address this comment and the previous ones about the ATL06 segments snow classification we reformulated the whole paragraph as follows and move it in a dedicated section:*

*"**4.1. ATL06 snow-cover calculation***

*The number of photons used to calculate the height of each ATL06 segment (n_fit_photons) varies upon the land cover. In particular, it increases over snow surfaces which are highly reflective in the ATLAS beam wavelength (532 nm). We take advantage of this property to determine the snow presence for every segment. We classified snow as present when n_fit_photons exceeded a certain threshold. We determined the threshold by optimizing the accuracy of the classification in comparison to MODIS snow cover data. We generated a daily gap-free stack of MODIS snow cover area maps by linear interpolation of the normalized difference snow index (NDSI) in the time dimension on a pixel basis followed by a binarization to snow and no-snow using a NDSI threshold of 0.2 (Gascoin et al. 2022).*

*Then, we sampled the MODIS snow maps at each ATL06 segment location for the matching date. The kappa coefficient, a statistic often used to measure the consistency between two classifications (Cohen, 1969), was used to find the optimal threshold, i.e. we determined the threshold which maximized the kappa value by testing all possible values from 0 to 500 photons (Figure S3). This optimization was done separately for the weak (N=123513) and the strong beam segments (N=132289). Figure S1 shows the spatial distribution of the mean annual snow cover duration computed from the interpolated MODIS snow maps over the Tuolumne river basin.”*

P3, L96ff: See also main comment. Why 15 m (20/40m may be more logical)? A map with snow-off (and snow-on) points would be useful. Is the point2dem utility meant for a dataset of profiles with that many gaps in between?

*Following another comment, we eventually excluded the snow-off DEM calculated from ATL06 snow-off points. 15 m was chosen for several reasons:*

*“The window used to select the photons and calculate the ATL06 elevations has a maximum length of 40 m and a width corresponding to the ATLAS footprint that is 11 m. The ASO and Pléiades DEMs were resampled from their native resolution of 3 m to 15 m by averaging. The 15 m resolution was selected because (i) it approximates the spatial window used to calculate each ATL06 segment and (ii) it is a multiple of the initial resolution of the source DEMs.”*

P4, L103ff: What do you mean by "extracted"? Filled with miscellaneous external products: is this the case for your study area?

*We rephrased:*

*“A DEM was clipped from the Copernicus-30 global dataset at its native grid spacing of 30 m.”*

*After verification, the Copernicus DEM is from the World DEM, Tandem-X in this area:*

*“The Copernicus-30 product is derived from InSAR data of the TanDEM-X mission in most areas, including the upper Tuolumne basin, with some areas filled with miscellaneous external products.”*

P4, L107: "the exact spatial scale of the ATL06 points remains uncertain": what do you mean? Please explain.

*We eventually deleted this part as the snow depth at the highest resolution were removed.*

P4, L122: Did you harmonise the vertical reference systems of the datasets? Please mention this step in the processing since a universal vertical shift will not reconcile elevation differences due to different ellipsoids/geoids for larger areas, and users of your approach should be aware of this. The high values for the Copernicus DEM in the table in the appendix suggest that this dataset may be based on a different ellipsoid or geoid?

*Good point, we neglected that. The Copernicus DEM is provided as height above the EGM*

*2008 geoid while the other datasets are height above the ellipsoid WGS84. We now corrected the Copernicus DEM to be height above the ellipsoid.*
*This modified the value of the co-registration vector, especially the vertical component which is down to -0.80 m (mode of the residual distribution).*

P5, Table 1: Note that the spatial resolution of the ATL06 data is 40 m, not 20. If included at all in this table, this dataset may need to be its own category (separated by a line?) given that it is not gridded and includes both snow-on and snow-off data. Also, the generated ICESat-2 snow-off DEM is missing in this table.

*We changed the name of the column to "Spatial spacing" to remain accurate.*
*We keep separated by a line the elevation dataset from the snow depth datasets but added a column saying if the dataset contains snow-on and/or snow-off data.*
*The ICESat-2 DEM was indeed missing but is not used anymore and thus was not added to the table.*

P5, L140ff: Is this really true for photon-counting lasers? ICESat was a full waveform sensor which is a very different system. The ICESat-2 ATL03 (photon) product has a spatial positioning uncertainty of ca. 5 m for individual photons, and the along-track resolution is only 0.7 m - and in the case of snow, there are often several photon returns for each shot (i.e. several per 0.7 m). The resulting photon density represents terrain variations one order of magnitude smaller than the spatial resolution of the ATL06 product.

*It is true that it is less obvious that slope is a source of errors for ICESat-2 than ICESat. We verified this by plotting the along-track slope and the vertical error estimated by the ATL06 algorithm and provided as the fields dh_fit_dx, sigma_h_mean (figure below added in the supplement). Empirically, the error seems stable for slopes up to 60° and increases for steeper slopes. We added:*
*"Thanks to the spatially dense photon detection of ICESat-2, the uncertainty of ATL06 only increases for slope greater than 60°"*

[Figure]

**Figure S2.** *Vertical uncertainty of the ATL06 elevation (sigma_h_mean) against along-track slope (dh_fit_dx). The along-track slope (dh_fit_dx) is directly provided in ATL06.*

P6, L147ff: What does this dataset contain? Land cover, forest..? Could be introduced together with the MODIS snow dataset.

*We now describe these dataset:*
*"The Terra MODIS MOD10A1 product was used to retrieve snow cover (Hall and Riggs, 2016; Figure S1). It provides daily snow cover maps with a spatial resolution of approximately 500 m (Hall and Riggs 2016). The tree cover density was retrieved from the Landsat-MODIS product (Sexton et al., 2013) which provides the proportion of the area occupied by trees at 30 m resolution (Figure 1)."*

Results

--

P6, Fig. 1: This is the only figure that shows snow depths - for only one of the studied tracks. From the title and abstract I would have expected more.

*We are not really sure what is misleading in the title and would need details on your thoughts to improve it. In the abstract as well, the results mentioned are about the residuals statistics in agreement with the focus of the article. Following this remark, we added a transect of snow depth as Figure 4.*

a) this map is not enough to characterize the study site (but would work for introducing this particular overpass if there was a proper study site map, see comment above). Why are the ICESat-2 tracks cut in the North and West? What is the extent of the snow-off DEMs?

*We created a figure which shows the extent of the dataset (Figure 1). The ICESat-2 tracks are cut due to the definition of the area of interest used to produce ATL06 (i.e. rectangular buffer around glacierized areas).*

b) what is the regression line of this plot? It does not seem to be 1:1

*In case you were asking what is the black line: it is the 1:1 line, we now labeled it. In case you were asking what is the linear regression of this dataset, we calculated it and added it to the plot. For the latter case, the coefficient of the linear regression is 0.91.*

c) axes labels? Does this include only data points (grid cells) where you have data from either dataset? There seems to be a lot of snow-free data points in the ASO data, does that mean that these were classified as snow-covered in your ICESat-2 dataset? Is this plausible?

*We added axes labels to the histograms. It includes the IS2-ASO 3 m snow depths and the ASO snow depths at the segment's position. The snow-free points in the ASO data are plausible as it might result from error in classification of the snow cover in our product, in ASO or from differences in the classification calculation in both methods. Unfortunately, little description of the ASO snow cover calculation is provided in Painter et al. (2016).*

d) axes labels? It would be useful to label median and NMAD (or accuracy and precision / bias and uncertainty). How does the distribution of the residuals of the snow-free ICESat-2 data points look like in this area or for the same overpass?

*We added lines for the median, plus/minus the NMAD and the snow-free data histogram. The exact values of the median and NMAD are provided in Table S2.*

P6, L159f: the figure in the appendix doesn't tell how robust the photon threshold is.

*The figure shows a clear unique optimum from about 130e3 segments in both cases (weak and strong beams) which is why we wrote "robust". However we agree that this term is vague hence we rephrased to :*

*"The optimization of the photon count threshold gives a clear and unique optimum (Figure S1) with 50 photons for the weak beam points and 186 photons for the strong beam segments."*

P7, L165ff: Are these numbers for snow-on or snow-off data points?

*We now precise:*

*"Some snow-on and snow-off points were obtained in areas with higher tree cover density up to 70%, close to the maximum observed in the upper Tuolumne basin (72%)."*

P7, Fig. 2: the labels for the panels are swapped. What do the vertical histograms actually show - the distributions of the number of photons contributing to all the different ATL06 data points? Please add this in the caption. What unit is the MODIS snow cover area?

*We updated the figure accordingly.*

P7, Fig. 3: The data in this figure is barely readable as the very narrow data stripes disappear, both digitally and in print. The elevation colour scale is indistinguishable with the dark grey background. If you want to show the location of the snow-off points, it may be better to provide a scatter plot (with larger markers). Coordinates, map layers etc. missing.

*We agree, we removed this figure.*

P8, L178ff: Readers might still think 12 days is rather long given the large variations in elevation/aspect/vegetation cover in the study area. You could provide met station time series in the supplement.

*We already comment on the evolution of the snowpack at the station:*

*"The snowpack changed a little as the Lower Kibbie Ridge station (2042 m a.s.l., 10 km east of the basin) measured +0.01 m water equivalent (w.e.) accumulation between the ICESat-2 track (12 March) and the ASO snow depth map (24 March) (SNOTEL data)."*

P8, L184ff: Are you comparing the same data points for all datasets, i.e., only the area

where you also have the Pléiades reference DEM? Otherwise the bias/precision values may not be directly comparable and could be an artefact of the different spatial sampling with different terrain/forest/elevation characteristics.

*The set of points are different for both datasets, we now insist on that:*

*"More points were available for IS2-ASO (N=5449) than IS2-Pléiades (N=1295), making the evaluation more robust for the former, but also possibly impacting differently the uncertainties of each methods on that date (see 5.3 and 5.4)."*

P8, L188: The figures in the supplement indicate that the poor performance is caused by a persistent (and spatially consistent?) spatial shift that looks like it could be removable with the Nuth/Kääb approach (?). Give it one more try with a tiling approach? Likely, many readers would have greatly appreciated if this approach worked with the COP30 DEM, as many readers might not have a snow-free DEM of ASO quality available.

*This results changed after we found a bug in our co-registration routine. We were inconsistently using the corner and the center of the pixel coordinate which produced an artificial shift of the data DEM. With this correction, the co-registration of Copernicus 30 m to ICESat-2 snow-off is similar to Copernicus to ASO DEM. Thus we deleted this part.*

*We repeated the tiling approach described above for the ASO DEM (see answer above).*

P8, L205: It would be interesting to discuss the slope-dependance of the performance of the different datasets - especially because it differs for the different datasets (increasingly positive bias for ASO, increasingly negative for Pléiades. I could not find this in the discussion section). Any ideas why it is different for these two datasets?

*We now insist on this slope effect but are unfortunately unable to explain it:*

*"The lidar airborne and satellite snow depth uncertainties differ largely in slopes with the increase of the bias for IS2-ASO with slope compared to the constant bias for IS2-Pléiades (Figure 5a). This discrepancy between the two DEMs is observed as well even when they are co-registered together (Figure S2 in Deschamps-Berger et al., 2020) but remains unexplained."*

P9, L215f: The mode requires binning of the data. What is the bin spacing in your case? Coarse bin spacing may result in remaining vertical shifts/bias.

*We used a binning of 5 cm.*

P9, L226ff: you introduce several new parameters and an entirely new processing aspect here, and it's difficult to understand why you did this and what effect it would have had on the results. This should probably be moved to the methods section (and the parameters properly explained).

*We now only present the results obtained with h_li. We added explanations in the data*

*section:*
*"3.1 ICESat-2 ATL06 elevation product*
*[...]The height h_li was used as it is calculated after correction of h_mean for errors in the detection of photons by ATLAS (i.e the transmit-pulse-shape error and the first-photon-bias)."*

P10, Fig. 4: Why are some boxplot classes missing in some plots? E.g., b) IC2-IC2 has only five boxplots.

*We added the explanation in the legend of the figure:*
*"Note that the sampling of the breakdown variables differs due to the different coverage of the snow-off DEMs."*

You chose to plot the individual data sets according to the delay between ICESat-2 and ASO acquisition, but there seems to be no detectable signal/dependency on the delay, and this is nowhere discussed. Personally, I would have preferred a timeline or any other way of arranging the boxplots that shows the acquisition dates, as the timing within the snow season provides a lot of meta information - it affects which elevations (and thus vegetation cover, slopes etc.) are included, and whether this is during the accumulation or melting season with their rather different snow pack.

*As suggested, this could be plotted in many different ways. While we understand the benefit of a temporal plot, we think that showing the lack of impact of the delay between ICESat-2 and ASO is an interesting result which consolidates the evaluation approach.*
*We added a comment on that in the results:*
*"No trend in the snow depth residual with the delay between ICESat-2 and ASO acquisition date is detected (Figure 5 d, e)."*

L240: You wrote earlier that the Pléiades DEM is not covering the entire study area?

*This sentence was indeed confusing, we deleted it as the IS2-IS2 product has been removed.*

Discussion

--

P11, L260: What leads you to the conclusion that the reason for your unsuccessful coregistration is the "coarse" pixel size of 30m? The spatial resolution of the dataset should not matter for the coregistration approach you used (Nuth/Kääb with interpolated elevations) - you could test this with a deliberately shifted 30m version of the ASO DEM. I am convinced the reason for the poor performance lies somewhere else than in the spatial resolution of the dataset.

*This result was modified by the correction of the co-registration routine. The co-registration vector of the Copernicus DEM is actually small and should not be the cause of the large error on snow depth.*

P11, L267f: "This approach could not be applied in the general case" - unclear, what do you mean here?

*We meant that co-registering the Copernicus DEM to the ASO DEM requires an airborne lidar DEM which is often not available in other study sites. This sentence was deleted as this result is no longer valid (see previous answer).*

What do you mean by "sufficient accuracy"? Resolution? Or would the data be better in Europe? How about testing other US/global DEMs (e.g., the SRTM DEM)?

*We deleted this sentence.*
*We leave the comparison to more snow-off datasets for future work (SRTM...).*

P12, L279: Shean et al. (2022) is missing in the reference list

*We added the reference:*
*Shean, D., Swinski, J. P., Smith, B., Sutterley, T., Ugarte, C., Lidwa, E., and Neumann, T.: SlideRule: Enabling rapid, scalable , open science for the NASA ICESat-2 mission and beyond. Journal of Open Source Software, 8, 1–6. https://doi.org/10.21105/joss.04982, 2023.*

P12, L288ff: This analysis shows that the ICESat-2 tracks are not representative for your particular catchment (which is fairly large). But are the ICESat-2 data points accurately reproducing the snow depths/elevation gradient of the corresponding locations/pixels in the ASO map? If yes, then the ICESat-2 data may very well allow an accurate estimation of the snow volume in the catchment, given a smart spatial extrapolation.

*Yes, we agree with this comment. We clarified this part:*

*"The ICESat-2 track only covers parts of the elevation with snow cover, and the snow depth distribution **sometimes** differs in both dataset over the sampled altitudes. **Estimation of the snow volume in a basin from ICESat-2 data requires to overcome the spatially discontinuous and variable sampling of ICESat-2 for instance through extrapolation based on topographical variables (Molotch et al., 2005, McGrath et al., 2018) or through data assimilation (Magnusson et al., 2014, Cluzet et al., 2022).** Another promising approach to utilizing ICESat-2-derived snow depth transects comes from Pflug and Lundquist (2020), where snow patterns in the upper Tuolumne basin were shown to be repeating and scalable. Small strips of snow depths were matched with a library of distributed snow depth maps from prior years to produce distributed snow depth maps of the basin. An ICESat-2 track might be used in this way to represent a relevant subset of a basin."*

P12, L296f: "Each ICESat-2 ATL06 snow depth point is informative over a small sampling area...": I disagree with this conclusion - see main comment about the resolution of the data. In addition, the snow depth uncertainty of individual ATL06 data points seems far too large for such a statement.

*We deleted this sentence.*

Conclusions and subsequent sections

--

P13, L310ff: This sentence is hard to understand (in particular because a "while" connects the two parts). Please rephrase.

*We rephrased:*
*"An enhanced tree cover degrades the snow depth derived from ICESat-2 ATL06 combined with a digital surface model (i.e. satellite photogrammetry) but has little impact if ATL06 is combined with a digital terrain model (e.g. from airborne lidar)."*

P13, L317: there is no author with initials NLM. JLM?

*Modified for JILM (Juan Ignacio Lopez-Moreno).*

P13, L335: this publication is from 2022, not 2021

*Modified.*

**Citation**: https://doi.org/10.5194/tc-2022-191-RC2

---

## Author Response (AR2)

Answer to comments (#2)

Evaluation of snow depth retrievals from ICESat-2

using airborne laser-scanning data

tc-2022-191

We thank the reviewers for their positive feedbacks on our revised version of the article. We took into account their comments as described below. We also included a number of stylistic and proofreading corrections from a co-author who is native English speaker.

Submitted on 21 Apr 2023 Anonymous referee #1

The authors have provided a manuscript that is strongly improved from the original iteration. I especially appreciate Section 6.1 - it is becoming clear from these ICESat-2 snow depth studies that accurate snow-off DEMs is necessary, and it is encouraging to see the results using Pleiades.

My comments at this point are minor and mainly highlight typos, but there are a few points worth addressing:

Page 1, Line 19: "taylored" --> "tailored" *Modified.*

Page 2, Lines 63-64: Suggested rephrasing: "ATL08 snow depth retrievals were found to be reasonably accurate in regions of low slope, but uncertainties increased in mountainous terrain, as previously found by Hu et al., (2021)." *Modified*.

Section 2: This section is fairly small. I suggest merging it with Section 3.

We added this section following a comment from the other reviewer suggesting to better structure the data, study site and methods sections. Instead of merging it with Section 3, we expanded it as follow:

«The upper Tuolumne river basin is part of the Sierra Nevada mountain range (California, USA) and is contained within Yosemite National Park (Figure 1). It is located above the Hetch Hetchy Reservoir which provides fresh water and produces hydropower for the San Francisco region (Painter et al., 2016). It consists of 1100 km2 of montane forests and alpine zones spanning an elevation range of 1200 m to 4200 m. Tree cover is composed of deciduous broadleaf and needleleaf evergreens forests and its density varies greatly within the watershed. More than half of the precipitation of this region range falls as snow (Li et al., 2017; Lahmers et al., 2022) with large year-to-year variations of snow accumulation related to low precipitation during pluriannual droughts or strong precipitation events from atmospheric rivers (Hedrick et al., 2019; Pflug et al., 2022).»

Page 4, Line 104: Extra period after "(4% of the data)". *Corrected.*

Page 4, Lines 104-105: The "n\_fit\_photons" variable is discussed in greater detail in Section 4, so I suggest removing this sentence. *Modified.*

Page 5, Line 119: Should be Section 3.3 (not 2.3). *Modified.*

Page 6, Line 148: "It can be used with \*a\* gridded product", or "It can be used with gridded product\*s\*"

Modified to «gridded products».

Figure 4: This is an interesting figure, though I suggest using shading or a bounding box to highlight the regions of high slope, rather than a black line.

We used a grey box to mark the regions of high slope and modified the caption accordingly.

Page 11, Line 221: Suggested rephrasing: "The IS2-ASO derived snow depths have a median bias of 0.00 m and a precision (NMAD) of 1.00 m."

We modified to:

«The IS2-ASO derived snow depths have a median bias of 0.00 m and a precision of 1.00 m (NMAD).»

Page 11, Line 235: 1.47 \*m\* *Corrected.*

Page 14, Line 288: "...even with tree densities up to 60%..." *Modified line 278.*

Page 14, Line 281: Suggested rephrasing: "Snow depths derived using the satellite photogrammetry DEM degrade rapidly when tree cover density increases and leads to marked bias." *Modified.*

Page 17, Line 381: snow depth\*s\* *Corrected.*

Report #2 Submitted on 22 May 2023 Referee #2: Désirée Treichler

The authors did a thorough revision where they addressed all reviewer comments in a satisfying way. My only regret is that the authors didn't choose to correctly map the actual footprint of the 40m long ATL06 segments: it could have been very useful for the readers to see whether that yields better results than the simpler/faster approach of using the interpolated value of the DEM at 15m resolution, as presented in this study. Now this question stays open.

I only have a few technical correction suggestions that don't require another review.

Minor comments

Abstract, L16/L19: 0.5m or 1m? Following the review discussion, consider whether the readers will correctly understand "precision" here or whether you should describe what you mean by it. These terms are later introduced in the article but not in the abstract, where they are already used.

We deleted «0.5 m» which was forgotten here and added the term «random error»: «However, using airborne lidar elevation model as snow-off elevation source yielded an accuracy of ~0.2 m (bias), a precision of ~1 m (**random error**) across the basin and an improved precision of 0.5 m for low sopes, compared to eight reference airborne lidar snow depth maps.»

Introduction, L68: I still suggest you write "...(ATL06) or \_land surface\_ and forest canopy height...". If "land surface" is not included here, it is not logical/unclear how the ATL08 product could be used for snow depths as referred to at L75ff. *Modified.*

Discussion, L424ff: The double coregistration in this part can be a bit confusing for the reader. I suggest you help the reader to keep track on which coregistration round you are on, e.g.: The residual shift of this second coregistration is small with respect to the DEM resolutions of ... highlights the good agreement of the original co-registration to the ICESat-2 data.

Vertical coregistration shifts (L427): of the first or the second round? Clarify in the text.

**We modified this part to clarify:**

«We co-registered the Copernicus DEM and the Pléiades DEM to the ASO DEM after **the original** co-registration to the ATL06 snow-off points to evaluate the success of the coregistration processes (Table S3). The small **horizontal shift obtained with the second co-registration**, with respect to the DEMs resolution, of 0.70 m and 1.38 m respectively for the Pléiades and the Copernicus DEM highlights the good relative agreement of the **original** co-registration. The **second** vertical co-registration **vectors** were significant with 1.15 m for Pléiades DEM and -0.65 m for the Copernicus DEM and could lead, if applied, to changes in the accuracy of the snow depths.»

Figure 6 / caption: could be made easier to read - it is not immediately clear that these are four groups of two panels each, and "top" and "shown below" (in the caption) are thus a bit unclear terms. You may want to add some more white space between the panel groups and/or label panels (a,b,c...) you can refer to in the caption.

We added a title, modified the y-label (SCA (km2)) and added a.,b... to improve readibility.

Colours in Fig. 3 and 6: Red/green with very similar intensity is an unfortunate choice (color-blindness, greyscale printouts). For example, red/blue with different intensity would be better readable.

We modified the line style to distinguish between these two colors.

Snow depth and snow cover area (SCA) distribution with elevation at four dates